# HIPEC in Ovarian Cancer Is the Future… and Always Will Be? Results from a Spanish Multicentric Survey

**DOI:** 10.3390/cancers15133481

**Published:** 2023-07-04

**Authors:** Alida González Gil, Álvaro Cerezuela Fernández-de Palencia, Álvaro Jesús Gómez Ruiz, Elena Gil Gómez, Francisco López Hernández, Aníbal Nieto Ruiz, Jerónimo Martínez, Iván Marhuenda, Pedro Antonio Cascales Campos

**Affiliations:** 1Departamento de Cirugía, Unidad de Cirugía Oncológica Peritoneal, Hospital Clínico Universitario Virgen de la Arrixaca, IMIB-ARRIXACA, 30120 Murcia, Spain; alidagonzalezgil@gmail.com (A.G.G.); elenagilgomez@hotmail.com (E.G.G.); 2Servicio de Cirugía General y del Aparato Digestivo, Hospital Clínico Universitario Virgen de la Arrixaca, IMIB-ARRIXACA, 30120 Murcia, Spain; alvaro.cerezuela@gmail.com (Á.C.F.-d.P.); alvarojesus.gr@gmail.com (Á.J.G.R.); pacolopezh3@gmail.com (F.L.H.); ivanmarcas99@gmail.com (I.M.); 3Departamento de Ginecología y Obstetricia, Unidad de Ginecología Oncológica, Hospital Clínico Universitario Virgen de la Arrixaca, IMIB-ARRIXACA, 30120 Murcia, Spain; anibal.nieto@um.es; 4 Departamento de Oncología Médica, Hospital Clínico Universitario Virgen de la Arrixaca, IMIB-ARRIXACA, 30120 Murcia, Spain; jeronimo@seom.org

**Keywords:** HIPEC, ovarian cancer, peritoneal carcinomatosis, peritoneal metastases, intraperitoneal chemotherapy, cytoreductive surgery

## Abstract

**Simple Summary:**

Treatment with intraperitoneal hyperthermic chemotherapy (HIPEC) has been criticized in ovarian cancer due to the lack of prospective and randomized clinical trials demonstrating its efficacy in improving outcomes. However, the publication of the Dutch clinical trial five years ago has not changed the usual clinical practice within the Spanish peritoneal oncologic surgery group. In this paper we analyze and reflect on the future of this technique in patients with peritoneal dissemination of ovarian cancer.

**Abstract:**

Ovarian cancer is the leading cause of death due to gynecological tumors in the female population. Despite optimal first-line treatment, including cytoreduction and platinum-based systemic chemotherapy, recurrences are frequent. The use of hyperthermic intraperitoneal chemotherapy (HIPEC) has been criticized, especially because of the lack of randomized controlled trials (RCTs) with convincing results to support the use of HIPEC in patients with ovarian cancer with peritoneal dissemination. In 2018, the clinical trial published by Van Driel et al. reported improved outcomes in favor of HIPEC treatment with cisplatin. In this study, we conducted a national survey within the Spanish group of peritoneal surgical oncology (Grupo Español de Cirugía Oncológica Peritoneal, GECOP) to explore the impact of the results of this RCT on clinical practice. A total of 33 groups completed the survey. Routine clinical practice was not changed in 28 of the 33 groups (85%) based on the results of the Van Driel trial. Despite the results of this RCT, most groups considered that more RCTs are needed and that, in the future, HIPEC may become the standard of care. In conclusion, the results from RCTs evaluating HIPEC treatment in patients with ovarian cancer has not been transferred to clinical practice.

## 1. Introduction

Ovarian cancer is the leading cause of death due to gynecological tumors in the female population. Despite the efforts, it has not been possible to implement an effective protocol for its early diagnosis, which means that most patients are diagnosed in advanced stages when there is already peritoneal dissemination of the disease [1]. After diagnosis, the cornerstone in first-line treatment is a combination of cytoreductive surgery (CRS), with the ideal goal of achieving a complete macroscopic resection of the disease, and platinum-based systemic chemotherapy [2]. In patients in whom initial cytoreduction is not feasible, the use of neoadjuvant systemic chemotherapy (NACT) allows a high percentage of complete cytoreduction with acceptable and similar results to patients with initial surgery without neoadjuvant chemotherapy [3,4].

Standard first-line treatment currently includes radical surgery with the aim of achieving complete cytoreduction of the disease and platinum-based systemic chemotherapy. However, the use of bevacizumab and poly(ADP-ribose) polymerase (PARP) inhibitors have gained importance in the last decade in selected patients in this clinical setting [5,6]. After optimal first-line treatment, the recurrence rate in ovarian cancer is high and is often located in the abdominal cavity [7]. To improve control of the microscopic component of the disease, the use of hyperthermic intraperitoneal chemotherapy (HIPEC) allows treatment with high doses of the cytostatic, in a single administration. The hyperthermic condition, in addition to the cytotoxic effects on the tumor cell, potentiates the effects of the administered drug [8]. In 2018, the largest clinical trial in ovarian cancer evaluating HIPEC administration after interval debulking, OVHIPEC-1, was published. The main objective of this clinical trial was to analyze the influence of HIPEC treatment on patient survival and, secondarily, morbidity, mortality, and quality of life. The results showed that disease-free survival (DFS) and overall survival (OS) were better when HIPEC was administered and that HIPEC treatment did not change morbidity and mortality rates or quality of life parameters relative to the control group treated with the more cost-effective CRS alone [9,10,11].

The most important criticism that has been argued against the use of HIPEC in ovarian cancer with peritoneal dissemination has focused on the low levels of scientific evidence from prospective, randomized clinical trials (RCTs). However, three RCTs have now been published in patients recently diagnosed with ovarian cancer with peritoneal dissemination [9,12,13] and in all three there appears to be a benefit in favor of HIPEC in patients previously treated with NACT and subsequent interval CRS. Undoubtedly, the most important clinical trial is OVHIPEC-1, and after its publication, the discussion was reactivated regarding the real position that HIPEC could take in the treatment of ovarian cancer, especially in patients treated with NACT. Some clinical guidelines, such as the one published by the National Comprehensive Cancer Network (NCCN), suggest that HIPEC may be a treatment to consider in newly diagnosed patients treated with NACT. However, the gynecologic oncology community and surgeons treating peritoneal surface malignancies have not incorporated HIPEC into the treatment of these patients nor adapted the cytostatic regimen described by the OVHIPEC-1 trial to daily clinical practice.

The aim of this work was to know the impact that the publication of the results of the OVHIPEC-1 clinical trial has had on clinical practice within the Spanish peritoneal oncological surgery group (Grupo Español de Cirugía Oncológica Peritoneal, GECOP), and to analyze, from a critical point of view, the reasons that hinder the practical application of results from RCTs such as this one in ovarian cancer with peritoneal dissemination.

## 2. Materials and Methods

The GECOP includes a total of 40 Spanish centers that perform CRS and HIPEC procedures in the treatment of peritoneal surface malignancies. The GECOP was officially founded in 2007 and is included within the Spanish Society of Surgical Oncology (Sociedad Española de Oncología Quirúrgica, SEOQ).

To find out what impact the publication of the OVHIPEC-1 clinical trial had had on routine clinical practice in patients with peritoneal carcinomatosis of ovarian origin treated by interval debulking after NACT and HIPEC, the GECOP member groups were invited to respond to a survey sent by mail in December 2022. Each week, an email reminder was sent to all groups that had not yet responded to the survey. The time allowed to respond to the survey was one month from the date of the first mailing. This survey was designed by our local group in Google Forms format and included a total of 22 questions regarding the center, number of patients treated by CRS + HIPEC each year, percentage of patients treated with the diagnosis of ovarian cancer, professional relationship (between surgical oncologist, gynecologic oncologist, and medical oncologist) in decision-making related to the surgical procedure, and usual HIPEC treatment schedule. In addition, questions were included about modifications made after the publication of the OVHIPEC-1 results, future opinions regarding the positioning of HIPEC treatment in this clinical setting, and whether new clinical trials similar to OVHIPEC-1 are needed. Data collection and data management were performed in Microsoft Excel, and descriptive statistics of the responses were performed.

## 3. Results

Of the 40 groups that are part of GECOP, a total of 33 groups responded to the survey.

The general profile of the groups that integrate the GECOP is surgeons with a preferential dedication to oncological surgery in general, and especially to the treatment of peritoneal surface malignancies, and who work within the Spanish public health system, in a referral hospital for complex oncological pathologies. Each group performs about 20–60 CRS + HIPEC procedures each year, and ovarian cancer usually occupies less than 50% of CRS + HIPEC-related activity. Two-thirds of the groups treat patients with ovarian cancer after NACT and there is adequate collaboration and synchrony between the surgeon, gynecologist, oncologist, and medical oncologist, who are part of the multidisciplinary team. The detailed data can be seen in Table 1.

Overall, 18 of the responding groups used open technique HIPEC, with cisplatin monotherapy being the most frequently used drug. In five groups, HIPEC treatment is not contemplated in any of the ovarian cancer scenarios. Of the 28 responding groups using HIPEC in ovarian cancer patients, up to 13 different HIPEC treatment patterns were identified based on the type of cytostatic, intraperitoneal temperature, and perfusion time (Table 2).

Most groups have shown a favorable opinion that the HIPEC treatment schedule published in the OVHIPEC-1 trial should be the one used in clinical practice. However, a total of 28 (85%) groups have not modified their HIPEC guideline, after the publication of OVHIPEC-1, to adapt it to the published scheme. Five groups did modify the guideline. Of these five groups in which HIPEC treatment was modified, two incorporated HIPEC after CRS in the treatment of ovarian cancer after that publication. Of the 28 groups that did not modify their HIPEC treatment regimen, 11 of them believe that a change is not necessary. Most groups consider that more clinical trials are needed before HIPEC can be widely recommended in ovarian cancer. Overall, 23 groups (70%) believed that HIPEC will become a standard treatment for ovarian cancer in the future (Table 3).

## 4. Discussion

The results of our study demonstrate that, beyond the favorable opinion regarding HIPEC treatment in ovarian cancer, only a minority of the groups modified their daily clinical practice after the results of OVHIPEC-1. In addition, it has been shown that there is great variability within GECOP in the HIPEC schedules used, with a total of 13 different guidelines, as was also shown in a previous survey conducted in the same group [14].

There are currently published data from three phase III clinical trials that have evaluated the use of HIPEC after CRS for ovarian cancer with peritoneal metastases at diagnosis. The first of these, which is the clinical trial on which the survey conducted in this paper is based, was published in 2018 by van Driel et al. [9]. In this trial, a series of 245 patients previously treated with NACT were randomized to receive or not HIPEC after interval cytoreduction. A total of 122 patients received HIPEC after CRS versus 123 in which only surgery was performed, without HIPEC. The results demonstrated that, with a similar rate of postoperative complications and no changes in quality-of-life analyses between the two groups, treatment with HIPEC was associated with favorable disease-free survival results of +3.5 months and overall survival of +11.8 months in favor of the group that had been treated with HIPEC (Cisplatin 100 mg/m^2^ body surface area for 90 min at an intraperitoneal temperature of 40 degrees Celsius). The second clinical trial, published by our group [12], was prematurely suspended due to the low level of recruitment. This clinical trial included a series of 71 patients diagnosed with high-grade serous epithelial ovarian cancer with peritoneal dissemination. All patients were previously treated with NACT. Patients were randomized to receive or not HIPEC after CRS (Cisplatin 75 mg/m^2^ body surface area for 60 min at an intraperitoneal temperature of 42° degrees Celsius). A total of 35 patients were treated by CRS and HIPEC versus a group of 36 patients in whom only surgery was performed, without HIPEC. Although the differences observed were not statistically significant, due to the low sample size achieved based on the expected sample size, a clinically valuable difference was found, with an improvement in disease-free survival in favor of HIPEC of +6 months. Again, no difference was found between the two groups with respect to postoperative complications or reported quality of life. Finally, the Korean clinical trial published by LIM et al. [13] included a group of 184 patients with stage III or IV ovarian cancer. The group treated with HIPEC received Cisplatin 75 mg/m^2^ body surface area at an intraperitoneal temperature of 41.5 degrees Celsius, for 90 min. The particularity of this clinical trial is that it included newly diagnosed patients in whom systemic neoadjuvant chemotherapy had or had not been administered. While in the overall analysis no significant differences were found in disease-free survival, which was the main endpoint of the trial, the analysis by subgroups showed, in the same way to those previously described in the other two clinical trials, a significant improvement of +2 months in disease-free survival in favor of HIPEC.

We know that RCTs are at the top of the evidence scale and therefore generate strong levels of recommendation for daily clinical practice [15]. It seems reasonable to think that, after an RCT, this clinical practice would be strengthened or weakened by the published results. However, the modification of clinical practice based on the results of an RCT is an unusual practice, especially in the field of surgery [16]. In addition, one-fifth of surgical clinical trials must be stopped early, and of those that manage to run to completion, one-third will never be published, and it is difficult to contact the principal investigators to obtain valuable information that may be important for the type of patients included in their study [17]. Considering the difficulties in achieving the expected recruitment of patients, the costs associated with conducting an RCT coupled with the expected low impact of the results in routine clinical practice makes the implementation of new RCTs challenging due to the possibility of wasting valuable time and resources without any compensation for the patients.

Specifically, the main argument against treatment with HIPEC in patients with ovarian cancer has been the lack of RCTs with convincing results to support its use. The acceptance of the results of an RCT in patients treated with CRS + HIPEC has been variable. The results of RCTs in patients with carcinomatosis of colorectal origin have had a very evident impact on clinical practice, with great detriment to the use of oxaliplatin. Today, the use of oxaliplatin as an HIPEC drug for the treatment of peritoneal carcinomatosis from colon cancer, or for prophylaxis in patients considered at high risk for the development of peritoneal disease, has decreased dramatically [18,19,20]. The only phase III clinical trial with positive results in favor of HIPEC in the treatment of patients at high risk of developing peritoneal disease in patients with T4 colorectal cancer has been published by Arjona et al. [21] using Mitomycin C. The value of Mitomycin C as HIPEC treatment after cytoreduction in patients with peritoneal carcinomatosis is also currently being evaluated by the GECOP group [22].

Even with the criticisms of the OVHIPEC-1 trial, our group considered that the impact on routine clinical practice had been lower than expected within GECOP. In fact, based on the results of the survey, only five responding groups (15%) modified their previous HIPEC guideline based on the results of OVHIPEC-1. The other 28 groups (85%) continued to maintain their previous HIPEC treatment regimen. In addition, more than a dozen different HIPEC schemes have been identified for the same scenario. Surprisingly, in Spain, there are still many chemotherapy guidelines for HIPEC treatment, reflecting the large amount of work pending by GECOP. This circumstance is not new. In 2011, our group published a brief letter to the editor in which we highlighted that the lack of RCTs and the great heterogeneity of the variables related to HIPEC treatment constituted a problem when making a specific recommendation [23]. More than 10 years have passed, and we seem to have conclusive information in the literature [24], but, at least in Spain, no progress has been made in the generalization of a homogeneous and consensual protocol. This fact complicates the implementation of HIPEC treatment in patients with ovarian cancer. An effort to reduce this variability in clinical practice has been made in the Netherlands, for example, using DELPHI methodology, under the premise that the OVHIPEC-1 HIPEC guideline should be the one used [25].

The impact of the results of an RCT can be attenuated if we also consider the time elapsed between the study design and the publication of its results, as it is difficult to go back and modify the design based on future advances in the treatment of ovarian cancer. In the case of OVHIPEC-1, recruitment began in April 2007, 7.5 years before approval by the EMA (European Medicines Agency) and FDA (Food and Drug Administration) of Olaparib for patients with BRCA+ status (around 2014). On the other hand, the approval of the combination of Bevacizumab with Olaparib was based on the results of the phase III PAOLA-1 study, published in 2019, 12 years after the recruitment of OVHIPEC-1 was initiated [5]. In this new era of personalized cancer treatment, one of the most important difficulties in extrapolating the results of previously described published clinical trials is not whether HIPEC is useful or not, but what is its relevance in this era of PARP inhibitors. Analyses of the data from the van Driel clinical trial with respect to BRCA status and HRD yielded some surprising results: survival outcomes were not modified by the BRCA status of the patient, which opens another line of discussion that does not benefit the defenders of HIPEC treatment. It is also likely that, in the near future, HIPEC will face another hurdle with the development of immunotherapy in the treatment of advanced and recurrent ovarian cancer.

Even with all these limitations, we continue to demand more RCTs [26] whose results, even with an ideal design, without changes in the treatment schemes of a given pathology and even with strong results, could also be ignored. Once the results of an RCT are published, surgeons have a special facility to look for problems for each solution provided by the publication. The most frequent argument is that the methodology is inadequate, even when the protocol of an RCT has been previously evaluated by experts in research methodology from the ethics and clinical research committees of the centers where it had been developed, been studied by members of the scientific expert committees of the institutions or governments that financed the project—often as part of a competitive call for proposals, and been re-evaluated by peers and the editorial board of the very high impact journals in which the results were published.

Also particularly relevant is the fact that the vast majority of RCTs are not supported by the pharmaceutical industry [27]. In surgical clinical trials, recruitment is often slow and research costs are high. Many RCTs must be suspended due to the lack of patients included or because the therapeutic innovation of the pathology they are studying changes during their development, especially in oncology. Although clinical trials with a larger number of enrolled patients are more likely to be published, multicenter designs to achieve better recruitment do not seem to be the solution [28]. In addition, industry-sponsored RCTs are more likely to generate positive results than RCTs funded by public, non-commercial institutions [15]. These are conflicts of interest that must also be considered. Moreover, we should not forget that part of the scientific community that is strongly against HIPEC in ovarian cancer is indeed the one that has the easiest access to ovarian cancer patients, and by this we refer to the gynecologic oncology community. However, part of this community is making efforts for the visibility of HIPEC treatment in patients with ovarian cancer. A recent meta-analysis by Llueca et al. [29] has again shown that, in patients with advanced ovarian cancer previously treated with NACT, the addition of HIPEC to CRS demonstrates an improvement in DFS and OS, without an increase in the number of complications.

The main limitation of our work is that it is a cross-sectional study in which the opinions of the group leaders of the different Spanish GECOP centers were obtained. The clinical practice of these groups for the treatment of ovarian cancer is also influenced by hospital policy. However, we also consider that the results should be motivation to work on the development of a homogeneous guideline that will allow us to acquire more extensive experience regarding the results of HIPEC use in ovarian cancer.

## 5. Conclusions

In conclusion, based on the results obtained in this work, the authors believe that HIPEC in ovarian cancer is the future… and always will be?

## Figures and Tables

**Table 1 cancers-15-03481-t001:** Description of the groups that participated in the study.

Variables	*n*
Main activity in hospital (GECOP group leader)	
Surgery, PSM exclusively	18
Surgery, HBP surgery	4
Surgery, Upper GI	3
Surgery, Surgical Oncology	2
Surgery, Others	5
Gynecol Oncol	1
Type of hospital	
Public and reference for oncological complex surgeries	15
Public and not reference for oncological complex surgeries	8
Private	10
Hospital size (beds)	
>750	10
500–750	8
250–500	6
<250	9
CRS + HIPEC every year	
>60	3
40–60	5
20–40	17
<20	8
Patients diagnosed with ovarian cancer (%)	
>75	1
50–75	7
25–50	11
<25	14
CRS + HIPEC in ovarian cancer	
Up-front	11
Interval Cytoreductive Surgery	22
Role of Gynecologist Oncologist and Surgical Oncologist	
Gynecologist Oncologist leads the unit	8
Leadership is shared	7
Surgical oncology leads the unit	12
There is no cooperation	6
Position of the Medical Oncologist at CRS + HIPEC	
Collaborator and in favor of the CRS + HIPEC	17
Collaborator but NOT in favor	15
Non-cooperative	11

**Table 2 cancers-15-03481-t002:** Description of the HIPEC treatment variables used by the participating groups.

Variables	*n*
HIPEC TECHNIQUE	
Open	18
Closed	11
Closed-CO_2_	12
HIPEC DRUG IN OVARIAN CANCER	
Cisplatin alone	18
Paclitaxel alone	5
Cisplatin + Doxorubicin	2
Cisplatin + Paclitaxel	2
Oxaliplatin	1
No HIPEC in Ovarian Cancer	5
TREATMENT REGIMEN (*n* = 28 groups)	
Cisplatin 100 mg/m^2^, 43°, 60 min	3
Cisplatin 100 mg/m^2^, 42.5°, 90 min	2
Cisplatin 100 mg/m^2^, 42°, 60 min	3
Cisplatin 75 mg/m^2^, 42°, 90 min	3
Cisplatin 75 mg/m^2^, 42°, 60 min	3
Cisplatin 80 mg/m^2^, 42°, 60 min	2
Cisplatin 50 mg/m^2^, 42°, 60 min	2
Paclitaxel 120 mg/m^2^, 41–42°, 60 min	2
Paclitaxel mg/m^2^ *, 41–42°, 60 min	3
Cisplatin 80 mg/m^2^ + Paclitaxel 120 mg/m^2^, 42.5°, 60–90 min	2
Cisplatin 80 mg/m^2^ + Doxorubicin 15 mg/m^2^, 42.5°, 60 min	1
Cisplatin 50 mg/m^2^ + Doxorubicin 15 mg/m^2^, 42°, 90 min	1
Oxaliplatin 200 mg/m^2^ + Doxorubicin 30 mg/m^2^ 41.5–43°, 60 min	1

* For every 2 L of solution for perfusion.

**Table 3 cancers-15-03481-t003:** Summary of the answers given by the different groups participating in the survey.

Variable	*n*
Should the OVHIPEC-1 scheme be used?	
NO	1
YES	19
I have doubts	13
Has the publication of OVHIPEC-1 changed your clinical practice?	
NO	28
YES	5
What has changed the OVHIPEC-1? (*n* = 5 groups)	
The duration of treatment has been increased	2
The type of drug	1
HIPEC treatment has been included in Interval Cytoreductive Surgery	2
Would a change from your current protocol to OVHIPEC-1 be necessary? (*n* = 28 groups)	
NO	11
YES	17
Do you think more RCT are necessary?	
NO	6
YES	27
Do you think HIPEC will be a standard treatment in ovarian cancer?	
NO	10
YES	23

## Data Availability

To access the data, please contact the corresponding author: cascalescirugia@gmail.com.

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
