# Peer review of "HIPEC in Ovarian Cancer Is the Future… and Always Will Be? Results from a Spanish Multicentric Survey"

_cancers, 2023, doi:10.3390/cancers15133481_

Round 1

Reviewer 1 Report

Thanks for the opportunity to review the manuscript “HIPEC in ovarian cancer is the future….and always will be? Results from a Spanish multicentric survey” My comments are listed below:

1.       The authors present the results of a survey among Spanish Gynaecological/oncologic surgeons within a national network concerning implementation of HIPEC treatment after publication of a Dutch study showing efficacy of HIPEC in conjunction with surgery for primary ovarian cancer.

2.       33/40 (82%) responded to the questionnaire. Reminders are not mentioned.

3.       Some linguistic errors can be detected in the manuscript (actividad en hospital, etc.).

4.       It is obscure how the authors categorized Position of the Gynecologist Oncologist in CRS + HIPEC as protagonist, co-protagonist, collaborator or non-cooperative.

5.       The title citing the statement that HIPEC will always be the future for ovarian cancer patients is a bit naïve. Treatments are rapidly evolving and novel treatments might replace HIPEC.

6.       Finally, I believe the results of this survey are more of national interest without significant international bearings.

Please revise language.

Author Response

Point 1. 33/40 (82%) responded to the questionnaire. Reminders are not mentioned.

Response 1. An explanatory sentence has been included in the new version of the manuscript submitted for publication. We have detailed this aspect in the second paragraph of the "Material and methods" section, line 84.

Point 2. Some linguistic errors can be detected in the manuscript (actividad en hospital, etc.).

Response 2. Grammatical errors and mistakes in the translation of the article have been reviewed again in the new version of the manuscript submitted for publication. The authors apologise and thank the reviewer for his comments.

Point 3. It is obscure how the authors categorized Position of the Gynecologist Oncologist in CRS + HIPEC as protagonist, co-protagonist, collaborator or non-cooperative.

Response 3. The authors have no "obscure" intention in relation to the reviewer's comment on this particular point. GECOP is a Spanish group that includes those teams that treat peritoneal malignant disease using CRS + HIPEC. These groups may consist of gynaecologists only, gynaecologists and surgeons, or surgeons only. However, we have reviewed the manuscript and in the new version submitted for publication, a relevant change has been made to table 1 (section “Role of Gynecologist Oncologist and Surgical Oncologist”)

Point 4. The title citing the statement that HIPEC will always be the future for ovarian cancer patients is a bit naïve. Treatments are rapidly evolving and novel treatments might replace HIPEC. 

Response 4. We fully agree, but in the title, the final question should not be overlooked: "...and always will be?”. This final question raises doubts as to whether HIPEC will become a future option for several reasons, some of which the reviewer mentions: the low consideration for the adoption of clinical trial results by surgeons; the fact that ovarian cancer is so heterogeneous, also at the molecular level, which determines the large number of clinical trials that are being developed on this basis; the reality that gynaecological oncologists and surgeons continue to approach this issue from a point of view that goes beyond science. The meaning we have wanted to give to the title of this paper with this final question is the same meaning that Roy Calne used in relation to xenotransplantation, which, in his opinion, had always been a promising therapy, but left open the possibility that it could never be consolidated.
(Calne R. Xenografting--the future of transplantation, and always will be? Xenotransplantation. 2005 Jan;12(1):5-6. doi: 10.1111/j.1399-3089.2004.00190.x. PMID: 15598267.).

Point 5. Finally, I believe the results of this survey are more of national interest without significant international bearings.

Response 5. It is possible that the reviewer is in some ways right, but of course this paper can serve as a wake-up call to the international community regarding what is happening, at least in Spain. A reflection is needed on what the real situation is regarding the acceptance/rejection of HIPEC treatment, at least in the clinical setting considered in this RCT.

Reviewer 2 Report

The manuscript entitled “HIPEC in ovarian cancer is the future…and always will be?” was reviewed. The authors tried to investigate if the clinical practice changed after a specific RCT of advanced stage ovary cancer by questionnaire. The authors disclosed the status of HIPEC for ovary cancer in the country. The idea and result are very interesting, though not scientific. The authors may need to clarify some points.

Major concerns:

1.     Were the detailed indication or patient selection criteria of HIPEC for advanced stage ovary cancer included in questionnaire? Because the disease status and comorbidities, disease severity, past chemotherapy usages…etc., may also differ the consideration of using HIPEC or not as well as the choice of chemotherapeutic agents. How do doctor in charge answer only one regimen they use?

2.     Why all doctors in Spain use fixed dose without adjustment by patients’ body surface area?

3.     Do you think other non-RCTs should be abandoned after this RCT? Please provide other evidence that the non-RCTs protocol are inferior to this RCT, not only referring the study design as a RCT. Please discuss more detailed, specifically, and scientifically. The authors seemed to emphasize that everyone needs to follow a specific protocol of one published study just because this is a RCT design. Most of surgeons may not be convinced that it is a final gold standard in the universe after one study published in an excellent Journal. The surgeons might still adjust their treatment protocol according to the patient's condition and institute facility. Because they may judge what is the best choice in current stage for the patient under different settings they stay. In addition, different treatment protocols are also necessary for giving more evidence to improve the outcome HIPEC (another RCT?).

4.     In the first sentence of abstract or introduction. “Ovary is the MOST lethal gynecological malignancy” Please verify it.

Please modify the usage of English words – RandomiSe, FavoUr…etc.

Author Response

Point 1. Were the detailed indication or patient selection criteria of HIPEC for advanced stage ovary cancer included in questionnaire? Because the disease status and comorbidities, disease severity, past chemotherapy usages…etc., may also differ the consideration of using HIPEC or not as well as the choice of chemotherapeutic agents. How do doctor in charge answer only one regimen they use?

Response 1. This is a question that is relevant to the selection of ovarian cancer patients for CRS + HIPEC. However, in this paper, our intention was to explore how influential the van Driel clinical trial had been in daily clinical practice and in GECOP member groups for the clinical scenario studied. In the initial description of the survey sent out, it was clearly specified that the clinical profile of the patients asked about corresponded to those included in the clinical trial published by van Driel and which is the focus of the survey.

Point 2. Why all doctors in Spain use fixed dose without adjustment by patients’ body surface area?

Response 2. The reviewer is absolutely right and we apologize for this error, which has been corrected in the new edition of the manuscript submitted for publication. The doses used correspond to mg/m² body surface area (Table 2).

Point 3. Do you think other non-RCTs should be abandoned after this RCT? Please provide other evidence that the non-RCTs protocol are inferior to this RCT, not only referring the study design as a RCT. Please discuss more detailed, specifically, and scientifically. The authors seemed to emphasize that everyone needs to follow a specific protocol of one published study just because this is a RCT design. Most of surgeons may not be convinced that it is a final gold standard in the universe after one study published in an excellent Journal. The surgeons might still adjust their treatment protocol according to the patient's condition and institute facility. Because they may judge what is the best choice in current stage for the patient under different settings they stay. In addition, different treatment protocols are also necessary for giving more evidence to improve the outcome HIPEC (another RCT?).

Response 3. The authors partially agree with the editor's statements. First, we do NOT believe that the comparison of results between non-RCTs versus this RCT is reasonable. By no means should the clinical trials currently underway be abandoned, but frankly, we would expect more consideration for the reported results, at least in regard to the HIPEC regimen to be used. The clinical trial published by van Driel has shown benefits in favor of HIPEC even with the limitations that have been highlighted in the scientific community. Indeed, the scientific community has strongly criticized the lack of RCTs to support the use of HIPEC. Despite a positive RCT, the low adherence to its recommendations is striking. The Canadian guideline, NCNN, has incorporated, for example, HIPEC for the clinical scenario studied, at least as a suggestion.

Point 4. In the first sentence of abstract or introduction. “Ovary is the MOST lethal gynecological malignancy” Please verify it.

Response 4. Yes, is the leading cause of death due to gynecological tumors in the female population. For a better understanding, we have modified the first sentence of the introduction.

Reviewer 3 Report

I thank the authors to submit this interesting article which give a snapshot of the GECOP point of view. The GECOP is a well-know and active group in the field of peritoneal surface malignancies.

The article underlines a conflicting situation. On one hand, we have the OVHIPEC-1 RCT which has been conducted consistently and methodically and concluded in favor of interval CRS+HIPEC in patients with stage III epithelial ovarian cancer. OVHIPEC-1 demonstrated the benefit of an HIPEC regimen based on cisplatin at a dose of 100 mg per square meter and at a flow rate of 1 liter per minute was then initiated (with 50% of the dose perfused initially, 25% at 30 minutes, and 25% at 60 minutes).

And the other hand we continue to face robust scepticism despite evidence. And this behaviour lead to loss of chance for patients.

I note that others oncological specialities are more likely to accept evidence and to adapt their daily practice consequently. Based on RCT with less positive outcomes than those demonstrated by the OVHIPEC-1 RCT.

As rightly focused by the authors, surgeon's practices are also influenced by hospital policy locally. Health policies must be updated and comply the evidence advances.

Concerning the heterogenity of used HIPEC regimens highlighted by the authors, results from an international Delphi with more than 100 expert panelists coming soon. These should be helpful to reassure to adapt their practice complying the evidence.

Author Response

We thank the reviewer for his comments, with which we agree. As in politics, the data, which are the same for everyone, allow a multitude of interpretations that may even be contrary. We look forward to the results of the DELPHI study to which you refer.

Reviewer 4 Report

I read with great interest the manuscript, which falls within the aim of this Journal and offers a high-quality overview of the topic.

This article is very timely, since there is no robust evidences on the use of hyperthermic intraperitoneal chemotherapy for the treatment of ovarian cancer both after PDS and IDS.

Although the manuscript can be considered already of high quality, I would suggest to take into account the following minor recommendations:

- I suggest another round of language revision, in order to correct few typos and improve readability.

-One of the main important problems regarding ovarian cancers is the recurrence after surgery and first line chemotherapy. Usually, the recurrence of the disease poorly responds to first, and sometimes even second line chemotherapy. In this scenario, it is possible that the ovarian cancer inherent resistance may be due to reduced immunosurveillance and drug- resistant cells. Authors should discuss this hypothesis, referring to: PMID: 32518015.

-Accumulating evidence suggests that the management of ovarian cancer should be personalized taking into account the performance status of the patient, in particular in case of elderly women. It would be interesting to discuss this point of paramount importance, referring to: D’Oria O, Golia D’Augè T, Baiocco E, Vincenzoni C, Mancini A, Bruno V, et al. The role of preoperative frailty assessment in patients affected by gynecological cancer: a narrative review. Ital J Gynaecol Obstet. 2022 June p.p. 76-83 doi: 10.36129/jog.2022.34.

- The authors have not adequately highlighted the strengths and limitations of their study. I suggest better specifying these points.

Considered all these points, I think it could be of interest for the readers and, in my opinion, it deserves the priority to be published after minor revisions.

The whole text should be corrected by a native English speaker in order to make the work clearer and more readable.

Author Response

Point 1. I suggest another round of language revision, in order to correct few typos and improve readability.

Response 1. The reviewer is absolutely correct. A thorough revision has been performed and the typographical errors have been corrected in the new edition of the manuscript submitted for publication.

Point 2. One of the main important problems regarding ovarian cancers is the recurrence after surgery and first line chemotherapy. Usually, the recurrence of the disease poorly responds to first, and sometimes even second line chemotherapy. In this scenario, it is possible that the ovarian cancer inherent resistance may be due to reduced immunosurveillance and drug- resistant cells. Authors should discuss this hypothesis, referring to: PMID: 32518015.

Response 2. We agree with your comment. However, the clinical scenario on which we base the survey is initial surgery after neoadjuvant chemotherapy, and at this stage of the disease the possibility of platinum resistance is not yet defined. Interestingly, there is long-standing evidence that platinum concentration within platinum-resistant cells is higher under HIPEC conditions (Hettinga et al, PMID: 9192975). However, the speed in medical oncology is remarkable with new therapeutic targets and immunotherapy.

Point 3. Accumulating evidence suggests that the management of ovarian cancer should be personalized taking into account the performance status of the patient, in particular in case of elderly women. It would be interesting to discuss this point of paramount importance, referring to: D’Oria O, Golia D’Augè T, Baiocco E, Vincenzoni C, Mancini A, Bruno V, et al. The role of preoperative frailty assessment in patients affected by gynecological cancer: a narrative review. Ital J Gynaecol Obstet. 2022 June p.p. 76-83 doi: 10.36129/jog.2022.34.

Response 3. We thank the reviewer for the suggestion. However, the focus of this paper is not on patient selection, but specifically on how the results of a prospective, randomized clinical trial have impacted daily clinical practice for a particular clinical scenario of ovarian cancer. The authors have been very specific in presenting the survey to those responsible for the treatment of ovarian cancer with cytoreduction and HIPEC in their hospital so that they conform in their responses to the criteria used by van Driel in his RCT when answering.

Point 4. The authors have not adequately highlighted the strengths and limitations of their study. I suggest better specifying these points.

Response 4. The original manuscript includes the main limitations of the study. We believe that institutional policy, and the beliefs and interpretations of the working group, influence whether or not HIPEC is used in this setting.  As can be seen in the manuscript we submitted, with the same scientific information available, there are a multitude of "real life" options. We believe this is a troubling fact in all respects.